# A Numerical Investigation on the Collision Behavior of Unequal-Sized Micro-Nano Droplets

**DOI:** 10.3390/nano10091746

**Published:** 2020-09-03

**Authors:** Lijuan Qian, Jingqi Liu, Hongchuan Cong, Fang Zhou, Fubing Bao

**Affiliations:** College of Mechanical and Electrical Engineering, China Jiliang University, Hangzhou 310018, China; P1901085244@cjlu.edu.cn (J.L.); P1701085211@cjlu.edu.cn (H.C.); zhoufang8602@126.com (F.Z.); dingobao@cjlu.edu.cn (F.B.)

**Keywords:** unequal-sized droplet, binary droplet collisions, numerical simulation

## Abstract

Micro-nano droplet collisions are fundamental phenomena in the applications of nanocoating, nano spray, and microfluidics. Detailed investigations of the process of the droplet collisions under higher Weber are still lacking when compared with previous research studies under a low Weber number below 120. Collision dynamics of unequal-sized micro-nano droplets are simulated by a coupled level-set and volume of fluid (CLSVOF) method with adaptive mesh refinement (AMR). The effects of the size ratio (from 0.25 to 0.75) and different initial collision velocities on the head-on collision process of two unequal-sized droplets at We = 210 are studied. Complex droplets will form the filament structure and break up with satellite droplets under higher Weber. The filament structure is easier to disengage from the complex droplet as the size ratio increases. The surface energy converting from kinetic energy increases with the size ratio, which promotes a better spreading effect. When two droplets keep the constant relative velocity, the motion tendency of the droplets after the collision is mainly dominated by the large droplet. On one hand, compared with binary equal-sized droplet collisions, a hole-like structure can be observed more clearly since the initial velocity of a large droplet decreases in the deformation process of binary unequal-sized droplets. On the other hand, the rim spreads outward as the initial velocity of the larger droplet increases, which leads to its thickening.

## 1. Introduction

The collision dynamics of binary micro-scale liquid droplets play an important role in the development of nanocoating [1] and nano-spray technology [2,3,4]. Micro-nano droplet collisions commonly occur in microfluidics, which is widely used in bioscience, clinical medicine, and environmental science [5,6,7]. Prominent examples are the fuel-spraying process in an engine combustor and the efficient distribution of agricultural sprays and powder production processes in food and life sciences. The frequency and outcomes of droplet collisions in the combustion chamber will determine the size and spatial distribution of the droplet group, and, ultimately, affect the heat and mass transfer effect of the system [8]. The results of droplet collisions are widely used to improve product performance and industrial efficiency. For example, entrained droplets travel through the steam-water separator of the steam generator in a nuclear power station. Based on the dynamic behavior and the discipline of droplet collision in the steam, the results can be referred during the design and optimization of separation apparatuses since considering the factors affects the separation efficiency [9].

Scenarios of microscale and nanoscale droplet collision dynamics have been widely investigated by either experiments or numerical simulations. Nahla et al. [10] demonstrated a new approach for droplet coalescence in microfluidic channels based on selective surface energy alteration, which focused on the coalescence process of the micro-nano droplet. Lian et al. [11] experimentally investigated the bouncing dynamics of impact droplets on the superhydrophobic surfaces for shortening contact time, and established the relationship between Weber number and contact time to find the minimum Weber number that formed the pancake-like bouncing behavior. Tang et al. [12] developed a theoretical model based on energy balance to explain the size-ratio dependence of the Weber number for the unequal-sized micro-nano droplets collision. Experimental results showed that the transition Weber number between bouncing and permanent coalescence regimes weakly depended on the size ratio, while the transition Weber number that separated permanent coalescence and separation regimes increased significantly as the size-ratio increased. However, for binary unequal-sized droplets, they concentrated on the mode division of the collision complex, the boundary lines between collision outcomes, and the energy evolution under low Weber number. Therefore, the numerical simulation is used to analyze the collision process of micro-nano droplets such as the flow and the velocity fields at a relatively high Weber number.

For numerical investigations, Bardia et al. [13] used the volume-of-fluid method to examine the binary nanometer-sized droplets collision. It was shown that the collision process can be described as follows: a pre-impact phase that ends with the initial contact of both droplets, and a post-impact phase characterized by the merging, deformation, and coalescence of the droplets. Kim et al. [14] investigated the collision of polymer nanodroplets by molecular dynamics simulations. The results showed that initial kinetic energy was changed into potential energy of the droplets as elastic energy when they began to contact each other. Then, the kinetic energy was completely changed into the potential energy of the droplets when droplets reached maximum deformation. Nikolopoulos et al. [15] studied the effect of the Weber number on binary unequal-sized droplets’ collision at the size ratios of 0.5 and 0.6. It was found that the small droplet spread on the large one’s surface at a high Weber number, while the small droplet penetrated into the large one at a low Weber number. Kuan et al. [16] developed a parallel, adaptive Eulerian-Lagrangian interface-tracking method to investigate head-on collisions of water droplets at high Weber numbers. Their simulations found that the rim grew with bread-like structures, and the extent of the retraction and deformation on the rim becomes weaker as the Weber number increases. Nikolopoulos et al. [17] employed the volume of the fluid (VOF) method to investigate the effects of the size ratio on binary droplets’ collision in the reflexive regime. The collision phenomenon including the formation of filaments, the maximum deformation, the penetration of one droplet into the other, and the satellite droplet formation of binary droplets were discussed at the Weber number of 56.

In the above studies, investigations have been conducted by discussing the collision behavior of droplet, the effect of the size-ratio on droplet collision, and energy change for binary unequal-sized droplets at a low Weber number, and the collision dynamics of unequal-sized micro-nano droplets under the high Weber number (larger than 200) are important in many engineering operations even though they are not detailed now. The rim of the collision complex retracts and aggregates strongly due to surface tension at the Weber number range from 210 to 270, which causes complex deformation of the droplet. In this paper, deformation processes, the flow field, and the energy evolution of binary unequal-sized droplet collisions under a higher Weber number will be discussed in detail. The mode of the droplets’ breakup and the evolution of the filament structure with the trend of the rim expansion will be analyzed. It is also expected that this study will help identify the collision dynamics of binary unequal-sized droplets at We = 210.

This paper focuses on the application of the CLSVOF methodology for the numerical investigation of the collision behavior of unequal-sized micro-nano Droplets. This method can capture the complex evolutionary process of the interface with an accurate normal vector and good quality conservation, which has been applied successfully in the past work [18,19,20]. At the same time, adaptive mesh refinement (AMR) will also be used to reduce the computational expenditure.

The rest of the paper is organized as follows. Section 2 gives the introduction of the CLSVOF method, numerical model, and comparison with experimental results [21] at We = 210. In Section 3, the effects of the size ratio (from 0.25 to 0.75) and different initial collision velocities on the collision process of two unequal-sized droplets are studied in terms of the droplets’ deformation, the velocity fields, and streamlines as well as the energy evolution. The concluding remarks are given in Section 4.

## 2. Numerical Framework

### 2.1. Mathematical Formulation

Coupled Level-Set and Volume of Fluid method (CLSVOF) combines the advantages of the Volume of Fluid (VOF) method in mass conservation with the advantages of the Level-Set method in the accuracy of interface curvature. The main processes of tracking the two-phase fluid interface include initialization of the phase function, solving of the phase function governing the equation, solving of the phase function transport equation, and re-initialization of the level set function. In the present study, the CLSVOF is used to study the collision of the binary droplets, which is modeled as incompressible, isothermal, immiscible flow. The governing equations for continuity, momentum, and phase-fraction equations are shown below.
(1)∇⋅U=0
(2)$$\frac{\partial \rho U}{\partial t}+\nabla \cdot \left(\rho UU\right)=-\nabla p+\nabla \cdot \tau +F_{\sigma }$$
(3)$$\frac{\partial \alpha }{\partial t}+\nabla \cdot \left(\alpha U\right)=0$$
where **U** is the fluid velocity, *P* is the pressure, **F***_σ_* is the volumetric surface tension force, *ρ* is the mixture density and ***τ*** is the stress tensor defined as *τ* = −*µ*(∇**U** + ∇**U***^T^*), in which *µ* is the mixture viscosity. The mixture density *ρ* and viscosity *µ* adopting phase fraction *α* as a weight coefficient can be calculated as the transition of two-phase physical properties in the interface region.
(4)$$\rho =\alpha \rho_{l}+\left(1-\alpha \right)\rho_{g}$$
(5)$$\mu =\alpha \mu_{l}+\left(1-\alpha \right)\mu_{g}$$

In the original VOF function (Equation (3)), the additional compression term ∇ × (**U**_*c*_*α*(1 − *α*) is introduced to sharpen the interface [22]. An improved VOF equation can be written as:(6)$$\frac{\partial \alpha }{\partial t}+\nabla \cdot \left(\alpha U\right)+\nabla \cdot \left[U_{c}\alpha \left(1-\alpha \right)\right]=0$$
where **U**_*c*_ is the compression velocity to suppress diffusion of the interface.
(7)$$U_{c}=\min\left(c_{\alpha }\left|U\right|,\max\left(\left|U\right|\right)\right)\cdot \frac{\nabla \alpha }{\left|\nabla \alpha \right|}\;$$

In this case, *C_α_* is the compression coefficient, which determines the compressive magnitude. It is generally greater than or equal to 1. The term ∇*α*/|∇*α*| is introduced to represent the convection of volume fraction function normal to the interface. The term *α*(1 − *α*) can be used to ensure itself invalid outside of the interface area and the divergence terms ensure mass conservation in terms of the entire compression term.

Furthermore, the Level-Set (LS) function *ψ* is introduced to ensure the interface smoothness by calculating a smoother curvature. The phase-fraction *α* is solved by transport Equation (3) for the VOF function, and use Equation (8) as the prime guess value to initialize the LS field.
(8)Γ=0.75Δx
where Γ is a small non-dimensional number, which depends on the minimum mesh size Δ*x* around the interface, the calculation formula is as follows.

The LS function *ψ*_0_ can be re-distanced by a reinitialized equation.
(9)$$\frac{\partial \psi }{\partial \tau }=S\left(\psi_{0}\right)\left(1-\left|\nabla \psi \right|\right)$$
where ***τ*** is the artificial time step. *S*(*ψ*_0_) is a sign function defined as *S*(*ψ*_0_) = *ψ*_0_/|*ψ*_0_|. The solution of *ψ* converges to |∇*ψ*|= 1, and the interface position is defined as the contour-line of *ψ* = 0. The number of iterations (*ψ_corr_*) meets the following term.
(10)$$\psi_{corr}=\frac{\varepsilon }{\Delta \tau }\;$$
where *ε* is the interface thickness. The Level-Set function can be used to calculate the interface normal by **n** = ∇*ψ*/|∇*ψ*| accurately. The smoother curvature is derived by *k* = ∇ × **n**. Therefore, the volumetric surface tension force can be calculated below.

Where *σ* is the surface tension coefficient, and *δ* is the Dirac function used to confine the influence of the surface tension to a narrow area around the interface.
(11)$$\delta \left(\psi \right)=\left\{\begin{array}{ll} 0 & \left|\psi \right|>\varepsilon \\ \frac{1}{2\varepsilon }\left(1+\cos\left(\frac{\pi \psi }{\varepsilon }\right)\right) & \left|\psi \right|\leq \varepsilon \end{array}\right.$$

### 2.2. Numerical Model

The two spherical droplets are formed in a cubic computational domain, as shown in Figure 1. Pressure-outlet boundary conditions are applied to all the computational domain planes. *D_S_* and *D_L_* present the diameter of the small droplet and the large droplet, respectively. The distance (*L*) between two droplets is 0.5 mm. The initial impact velocities of the two droplets are *V_S_* and *V_L_*, and the relative velocity is defined as *V_r_* = *V_L_* − *V_S_*. In the present study, the computational domain size is 6.25 mm × 6.25 mm × 6.25 mm. The grid number is 50 × 50 × 50.

In the present study, the adaptive mesh refinement (AMR) is used, as shown in Figure 2. The minimum cell size Δ*x* is defined as *L*_D_/2^Lmax^, where *L*_D_ is the side length of a square, and *L*_max_ is the cell refinement level. The numerical resolution related to the cell refinement level of the droplet is computed as *D*_0_/Δ*x*, where *D*_0_ is the initial diameter of the droplet. The numerical simulation method and the grid independence are verified by Qian et al. [23].

We take *D_S_* as the characteristic length and the relative velocity *V_r_* of the impacting droplets as the characteristic velocity. Four dimensionless numbers are employed to describe the collision dynamics of binary droplets in this work: the Weber number (We), the Reynolds number (Re), the size ratio (Δ), and the dimensionless time (*T*). The Weber number (We) represents the ratio between inertial forces and the surface tension coefficient. The Reynolds number (Re) is defined as the ratio of the inertial and viscous forces. The size ratio (Δ) is the diameter ratio of the small droplet and the large droplet.
(12)$$\mathrm{We}=\rho_{l}V_{r}^{2}D_{S}/\sigma \;$$
(13)Re=ρlVrDS/μ
(14)$$\Delta =D_{S}/D_{L}$$
(15)$$T=(t-t_{0})V_{r}/((D_{S}+D_{L})/2)\;$$
where *ρ_l_* is the liquid density, *σ* is the surface tension coefficient, and *µ_l_* is the droplet fluid viscosity. In this paper, the liquid density *ρ_l_* is 1000 kg/m^3^, the surface tension *σ* is 0.073 N/m, and the viscosity *µ* is 0.001 Nm^−2^ × s^−1^. *t*_0_ is the initial contact time of two droplets. The parameters used in our simulation in this paper are shown in Table 1.

### 2.3. Comparison with Experiments

The evolution of the droplet collision deformation for the case of We = 210 is presented in Figure 3, which is compared with the experimental work of Pan et al. [21]. It can be observed that a circular sheet forms in the middle of the collision complex after the coalescence of the droplets. The circular sheet is radially extended within the collision plane and forms a circular rim at the fringe of the circular sheet due to the end-pinching effect of the surface force, which includes Taylor–Culick rim (TC rim) [24,25,26]. Compared to the simulation results, the process of the deformation matches well with the experimental work, and this numerical method can be used to investigate the binary unequal-sized droplets collision at We = 210. However, the deviation between experimental and simulated results is the extent of the retraction and deformation on the rim. The deformation of the rim was affected by the anti-symmetrical disturbance of the RT (Rayleigh-Taylor) instability and the symmetrical disturbance controlled by the RP (Rayleigh-Plateau) instability in experiments. To simplify the model, the initial disturbance that is equivalent to instability is not imposed in the numerical simulation.

## 3. Results and Discussion

### 3.1. The Effects of the Size Ratio (Δ)

To study the effect of the size-ratio on binary droplets’ collision, different size-ratios of 0.75, 0.50, and 0.25 (case A–C in Table 1) are considered under the same Weber number of 210.

When the size-ratio is 0.75 in Figure 4a, the small droplet spreads on the surface of the large one and flattens gradually at the contact surface as the rim expands outward radially. The circular rim expands outwards and ruptures with the circular sheet, evolving into the filament and satellite droplets around the complex droplet. Figure 4b shows that the small droplet attaches to the big droplet and penetrates deeply into the big one because of a pressure difference between them, which is in agreement with the work of Nikolopoulos et al. [15]. Then the complex droplet that forms a hole-like structure evolves into the band-like filament and breaks into satellite droplets at the later stage of collision. When the small droplet becomes smaller in Figure 4c, due to the low impact momentum of the small droplet, the small droplet spreads on the surface of the large one with shallower penetration. Then the rim evolves into four distinct petal-like structures, which forms the ellipse holes on the fringe of petal-like structures. Difference size-ratio collisions of unequal-sized mico-nano droplets show similar collision behavior that two droplets coalesce rapidly after the collision due to intermolecular forces and then deform on the axial and radial direction. However, the tendencies of the rim expanding during the collision processes are different for various size ratios.

Figure 5 shows the evolution of the axial and radial deformations of the droplets, *d_X_* is the maximum radial deformation, and *d_y_* is the axial deformation in which the dimensions are characteristic of the length of equivalent volume *D_eq_*, which is defined as $D_{eq}={\left(D_{S}^{3}+D_{L}^{3}\right)}^{1/3}$. *T* = 0 means the initial contact time of two droplets, as mentioned in Equation (17). It can be observed that the larger the size ratio, the smaller the axial deformation of droplets, and the larger the droplet spreads radially outward.

In order to further illustrate the effect of the size ratio on the droplets’ deformation, the streamlines and the velocity magnitude from the mixture at the cross-section Z = 0.003125 are given in Figure 6. It shows that, as the droplets move toward the collision center, gas is squeezed out, a gas jet sheet is formed between two droplets, and the streamlines form two opposite pairs of rotating vortexes at *T* = 0.00. As shown in Figure 6a, for binary equal-sized droplets collision, two opposite pairs of rotating vortexes are produced and attached to both sides of the rim. During the collision process, the droplets merge at a significantly flattened surface, and the rim expands outward with higher radial velocity while the axial velocity is dissipated gradually, which can be obtained in Figure 5. For binary unequal-sized droplets collision, as shown in Figure 6b–d, two different opposite pairs of rotating vortexes are produced due to the size ratio. When the size ratio is 0.75, when compared with binary equal-sized droplets collision, under the action of the opposite pair of rotating vortexes generated on the contact plane, a shallower hole-like structure is visible at *T* = 3.63 in Figure 6b. When the size ratio is 0.5 and 0.25, at *T* = 0.59, due to the detachment of the head of the main vortex by forming the opposite pair of rotating vortexes on the interface. The concave interface forms at the outer edge of the contact surface and is gradually stretched to the concaved crater at *T* = 1.62. The same stage of the formation of the main vortex by the vorticity generation from interfacial deformation was also observed and explained by Sun et al. [27]. Under the influence of collision dynamics, the vortex rings produced at the contact surface are dissipated at *T* = 3.63, and the hole-like structure is more clear, which is also shown in Figure 6c,d.

### 3.2. The Effects of the Initial Velocity

In order to illustrate the effect of the large droplet and the small droplet on the collision dynamics, the initial velocity of the droplets is changed under the condition of We = 210 and ∆ = 0.5, and the vector fields inside and around the colliding droplets are also further given, as shown in Figure 7 and Figure 8.

Figure 7 indicates that the momentum of the small and large droplets has different effects on the motion tendency of the mixing droplet and the expansion of the rim at We = 210. When the larger droplet impacts the smaller droplet with the velocity of 5.54 m/s, the mixing droplet integrally moves upward, forming the hole-like structure. Subsequently, the larger droplet continues to move and expand outward, which leads to a thicker rim structure. When the velocity of the smaller droplet is −5.54 m/s, the smaller droplet spreads only on the surface of the larger droplet and the motion of the mixing droplet remains almost unchanged. The mixing extent of the collision droplet is more distinct and the axial deformation of the collision droplet is larger. In Figure 8, the vector fields inside and around the colliding droplets are given to explain the flow direction of the small droplet and the large droplet. It is found that, when the droplets collide with the opposite initial velocity, the mass of the small droplet tends to flow into the large one, and spreads outward from the inside of the large droplet. In Figure 8b, when the velocity of the large droplet is 5.54 m/s, the large droplet with the large momentum deforms and carries the mass of the small droplet to spread outward widely. While the small droplet takes the velocity of −5.54 m/s, as shown in Figure 8c, a little mass of the small droplet spreads on the surface of the large drop with a shallower penetration. It is shown that, during the collision process, the momentum of the large droplet has a larger effect on the formation of the hole-like structure and the vector fields inside and around the colliding droplets also indicate that, for binary unequal-sized droplet collisions, the mass of the small droplet tends to flow into the large one after the collision, so that the colliding droplets are difficult to separate.

### 3.3. Evolution of Dimensionless Energy

In this section, following the research studies of Yoshino et al. [28] and Sarrokaet al. [29], the energy transfer processes are also discussed.

The initial kinetic and surface energy of the droplets at *T* = 0 are computed as follows.
(16)$$E_{S0}=\sigma S_{0}=\sigma \left[4\pi {\left(\frac{D_{S}}{2}\right)}^{2}\right]+\sigma \left[4\pi {\left(\frac{D_{L}}{2}\right)}^{2}\right]\;=\pi \sigma D_{L}^{2}(1+\Delta^{2}\;)\;$$
(17)$$E_{K0}=\frac{1}{2}\left[\frac{4}{3}\pi {\left(\frac{D_{S}}{2}\right)}^{3}\rho_{L}\right]V_{S}^{2}+\frac{1}{2}\left[\frac{4}{3}\pi {\left(\frac{D_{L}}{2}\right)}^{3}\rho_{L}\right]V_{L}^{2}\;=\frac{\pi \rho_{L}D_{L}^{3}}{12}(\Delta^{3}V_{S}^{2}+V_{L}^{2})\;$$

The kinetic and the surface energy at each time step during the collision process are computed as:(18)$$\begin{array}{l} E_{K}=\frac{1}{2}\sum_{ijk=1}^{N_{cells}}\left(f_{ijk}\rho v_{cell}\right)\left(u_{ijk}^{2}+v_{ijk}^{2}+w_{ijk}^{2}\right)\; \end{array}$$
(19)$$E_{S}=\sigma \sum_{p=1}^{N_{polygons}}SA_{p}\;$$
where *SA_p_* is the polygon area and υ_cell_ is the volume of the computational cell, *f_ijk_* is the volume fraction of the liquid phase, *u_ijk_*, *υ_ijk_*, and *w_ijk_* is the velocity component and *N_cell_* is the number of the computational cell.

The kinetic energy and surface energy are non-dimensionalized by the initial kinetic energy and surface energy, respectively. The evolutions of dimensionless energy of binary droplets under different size ratios and initial impact velocity are given in Figure 9.

Figure 9a,b shows that the larger size-ratio is, the larger the surface energy is, and the smaller the kinetic energy is. Corresponding to the maximum deformation process of droplets in Figure 4, it can be seen before deformation that the surface energy and the kinetic energy remain unchanged. As binary droplets move toward each other, the kinetic energy is transformed to the surface energy. The surface energy increases gradually with the kinetic energy decreasing. The larger the size ratio is, the more kinetic energy is converted to surface energy.

In Figure 9c,d, the energy evolution of the droplets with different initial velocity shows that the kinetic energy of the droplets with the opposite initial velocity is the largest, and the kinetic energy of the large droplet with the velocity of 5.54 m/s is larger than that of the small droplet with the velocity of −5.54 m/s. Compared with the other two cases, when the velocity of the small droplet is −5.54 m/s, it can be observed that, during the collision process, more kinetic energy is converted to surface energy. For the surface energy evolution of the droplets, it can be found that, when the velocity of the large droplet is 5.54 m/s, the surface energy is the largest. On one hand, the surface of the mixing droplet contracts as much as possible due to the surface force. On the other hand, the surface energy is increased by thickening the rim. Hence, the collision complex is stable without satellite droplets forming around the mixing droplet. The surface energy of the droplets with the opposite initial velocity is larger than that of the small droplet with the velocity of −5.54 m/s in the late stage of collision. Corresponding to the droplet deformation process, like Figure 7, it can be explained that, under the relatively high Weber number, the impact momentum of the small droplet has a great effect on the mass of the outward spreading rim, and the mass of the outward spreading rim decreases as the impact momentum of the small droplet increases. The collision dynamics of the droplets are mainly dominated by the large droplet, which had also been provided by Mohammad et al. [30] at a low Weber number.

## 4. Conclusions

Binary unequal-sized micro-nano droplets collision has been numerically simulated by a coupled level-set and volume of fluid (CLSVOF) method with adaptive mesh refinement (AMR). The effects of the size ratio and different initial collision velocities on the collision process of the droplets are discussed and summarized as follows.

For binary unequal-sized micro-nano droplet collisions, the maximum axial deformation of the complex droplet increases when the difference of the diameter between two droplets becomes smaller, and the complex droplet reaches its maximum value of radial deformation when ∆ = 1.The spreading effect of the collision complex becomes more clear as the size-ratio increases and conversely facilitates the mixing effect of the collision complex. As for the atomization combustion, the complex droplets contact the gaseous medium sufficiently when the collision droplets are equal-sized, which could promote the efficiency of combustion.When two droplets keep the constant relative velocity, the motion tendency of the droplets after the collision is mainly dominated by the large droplet. As for a steam separator, the velocity direction of complex droplets is maintained by increasing the velocity of large droplets due to the large momentum of large droplets. The complex droplets are hardly affected by the steam flow and much easier to remove by the wave-type vanes as a result.The mixing effect can be improved by suppressing the initial velocity of the larger droplet. Moreover, it is difficult to separate a droplet from another after the collision since the small droplets tend to penetrate into the large one.

## Figures and Tables

**Figure 1 nanomaterials-10-01746-f001:**
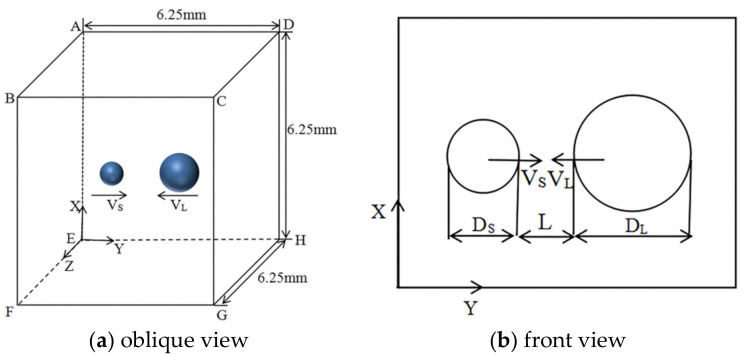
Schematic of the computational domain and the initial state of binary droplets. (**a**) the oblique view, and (**b**) the front view.

**Figure 2 nanomaterials-10-01746-f002:**
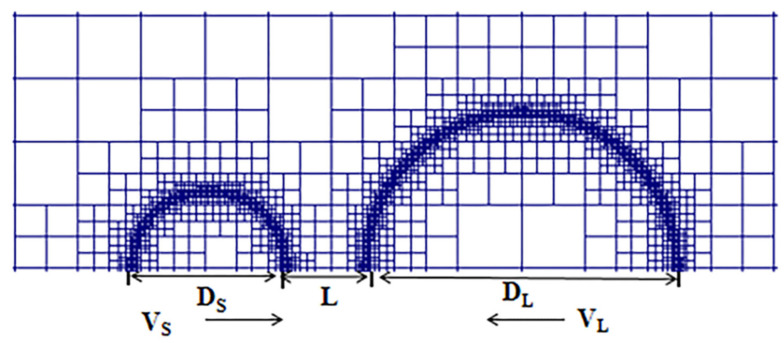
Initial droplets and adaptive mesh.

**Figure 3 nanomaterials-10-01746-f003:**
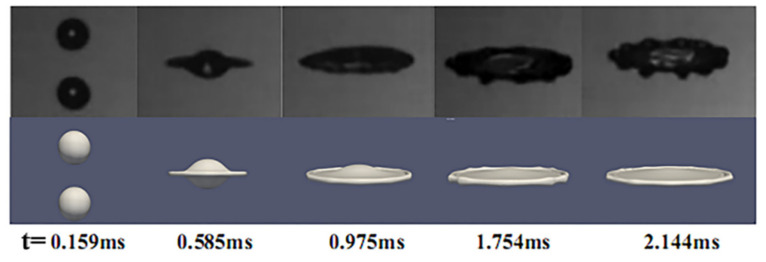
Experiments and numerical simulation results of binary equal-sized droplets collision with We = 210: Experimental result of Pan et al. (2009) [21] (above). Simulation result (below).

**Figure 4 nanomaterials-10-01746-f004:**
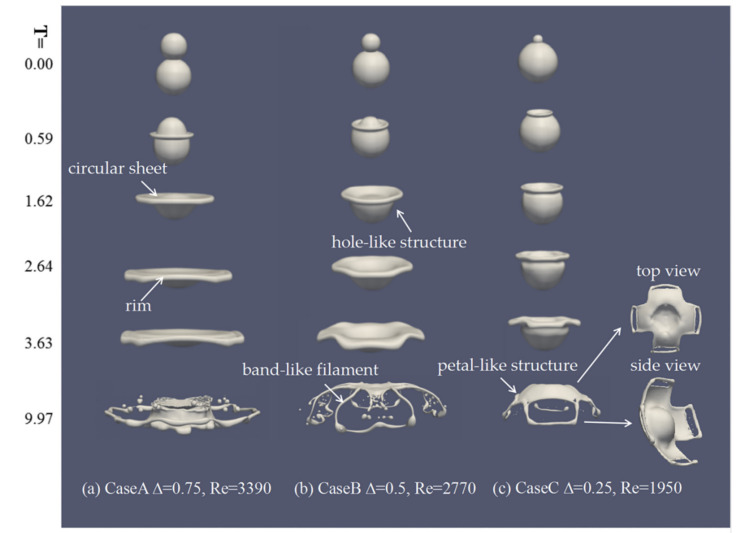
Temporal evolution of droplets’ deformation under the different size-ratios of 0.75, 0.50, and 0.25 at We = 210.

**Figure 5 nanomaterials-10-01746-f005:**
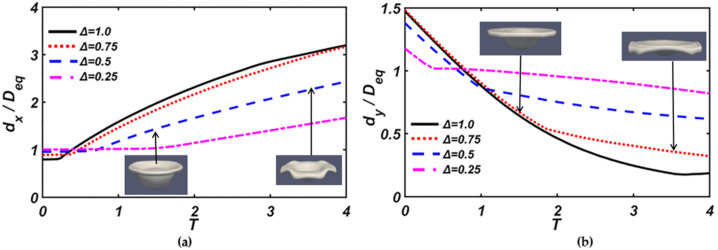
Evolution of the maximum diameter of the droplets with different size ratios: (**a**) the dimensionless maximum radial diameter of the droplet (*d_x_/D_eq_*), and (**b**) the dimensionless maximum axial diameter of the droplet (*d_y_/D_eq_*).

**Figure 6 nanomaterials-10-01746-f006:**
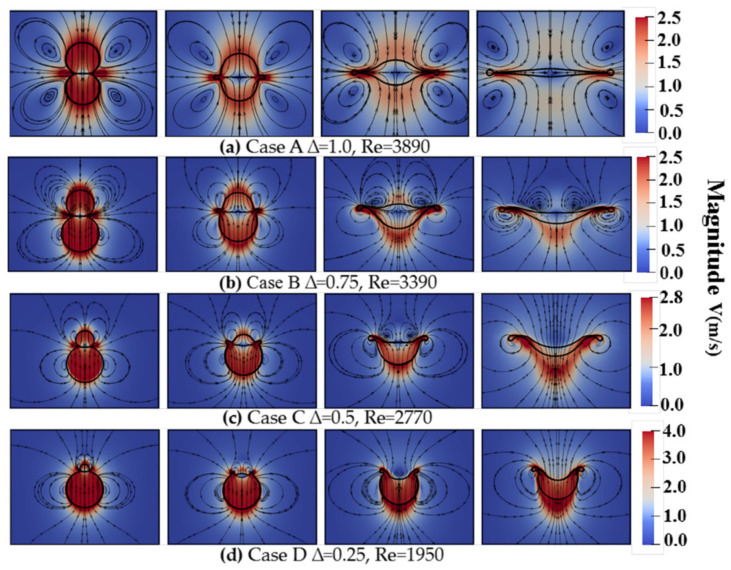
The streamlines and the velocity magnitude from the mixture at the cross-section Z = 0.003125 under the different size-ratios of 0.75, 0.50, and 0.25 at *T* = 0.00, 0.59, 1.62 and 3.63, respectively. (**a**) CaseA ∆ = 1.0, Re = 3890, (**b**) CaseB ∆ = 0.75, Re=3390, (**c**) CaseC ∆ = 0.5, Re=2770, (**d**) CaseD ∆ = 0.25, Re = 1950.

**Figure 7 nanomaterials-10-01746-f007:**
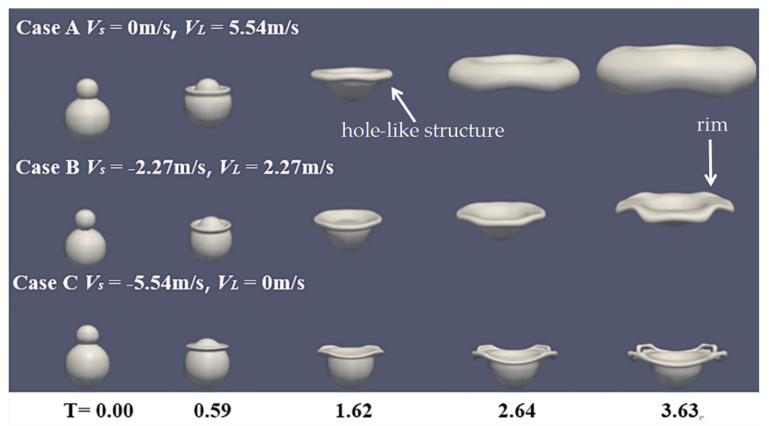
Shape evolution of binary droplets’ collision with the different initial velocity at We = 210, ∆ = 0.5.

**Figure 8 nanomaterials-10-01746-f008:**
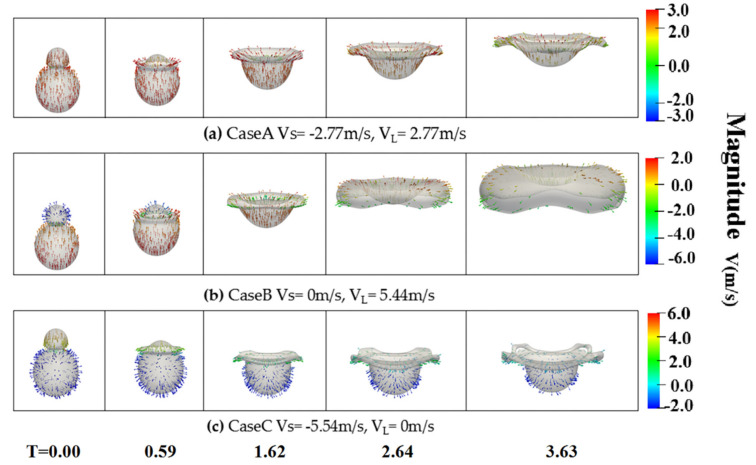
The vector fields inside and around the colliding droplets for the case of We = 210, ∆ = 0.5 at T = 0.00, 0.59, 1.62, 2.64, and 3.63, respectively. (**a**) CaseA *V_s_* = −2.27 m/s, *V_L_* = 2.27 m/s, (**b**) CaseB *V_s_* = 0 m/s, *V_L_* = 5.54 m/s, (**c**) CaseC *V_s_* = −5.54 m/s, *V_L_* = 0 m/s.

**Figure 9 nanomaterials-10-01746-f009:**
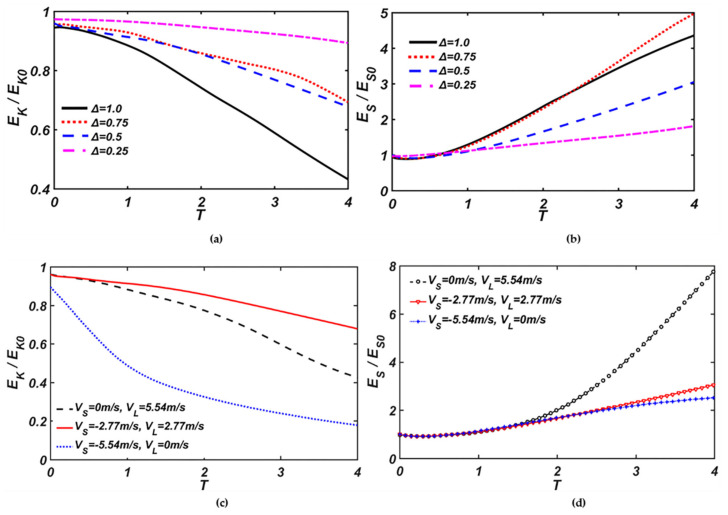
Evolution of dimensionless kinetic energy and surface energy of the droplets, the size ratio: (**a**) *E_k_/E*_*k*0_, (**b**) *E_s_*/*E_s_*_0_, the initial velocity: (**c**) *E_k_/E*_*k*0_, (**d**) *E_s_*/*E_s_*_0_.

**Table 1 nanomaterials-10-01746-t001:** Parameter settings in simulation cases.

Cases	We	Re	*V_S_*	*V_L_*	*V_r_* = *V_S_* − *V_L_*	∆ = *D_S_*/*D_L_*
A	210	3390	−2.26	2.26	4.52	0.75
B	210	2770	−2.27	2.27	5.54	0.5
C	210	1950	−3.9	3.9	7.8	0.25

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
