# Peer review of "A Numerical Investigation on the Collision Behavior of Unequal-Sized Micro-Nano Droplets"

_nanomaterials, 2020, doi:10.3390/nano10091746_

Round 1

Reviewer 1 Report

The manuscript presents a numerical model of the collision of binary droplets of different size ratios. The manuscript is well-written, the arguments are well-presented and quite clear. I would recommend the manuscript to be accepted with minor revisions. Here are my comments:

1. The authors presented the methodology and the potential applications for it. However, the novelty of the approach is not very clear. The authors should clearly state the novelty in the abstract and introduction sections. The authors should also state any new findings in these sections.

2. The unit for time axis is missing in the figures 4 and 7

3. The term "ligament" in Fig 4 should be replaced with "filament"

4. The figures showing the 3D view of the collision is very informative in Fig 4 and 7. However, to illustrate "petal" shape, it might be more helpful to show a top view in addition to the side view.

5. Fig 3 shows the comparison between a prior experimental work and the current simulated results. The author should include discussion on why the simulated results is qualitatively different from the experimental work, i.e. why is the wavy rim not present in the simulated results?

6. Fig 5 inset photos are too small

7. Fig 6 legend font size is too small

8. The conclusion is very clear and concise.

Reviewer 2 Report

Interesting and well written paper on an important phenomenon.

Remarks:

In the abstract size ratios of 0.25 up to 0.75 are mentioned in the paper also results for a size ratio equal to 1 are given.

108:  experimental and calculation results are pretty off for large t = 2.144 ms. Maybe the authors can explain, influence of surrounding air or different material properties.

302: data on right hand side of figure 4 are not specified probably T.

306: conclusion 1 is not clear, maybe larger should read smaller.

Reviewer 3 Report

The manuscript gives a numerical investigation of droplet collision. All the experiment look good and be can be accepted after minor revision. Only one question is In Figure 3, as authors cited figure from other work, please make sure you have obtained permission to reuse it.

Reviewer 4 Report

Paper Review

This paper presents the numerical study on the collision behavior of micro-nano droplets. In this study, a coupled level-set and volume of fluid method with adaptive mesh refinement is utilized to conduct the simulation. The effects of the size ratio and different initial collision velocities at We=210 are investigated. The results show that the surface energy converting from kinetic energy increases with size ratio which can increase the spreading effect. The structure of the droplet collision is also discussed. However, the motivation of this work is unclear and the novelty of this work is not well presented. Due to these reasons, I recommend the publication of this paper with revision.

The comments are as follows:

  • It seems this paper is a development of the previous work (Qian L, Cong H, Zhu C. A Numerical Investigation on the Collision Behavior of Polymer Droplets. J. Polymers. 2020, 12(2): 263.). The author may add one or two sentences in the introduction to highlight the novelty of the study.
  • Why set We=210? Are there any references to support or practical application? Would it be possible to vary We and how does it affect the result?
  • How does the result and discussion contribute to the industrial application?
  • Does the color of the vectors in Figure 8 refer to the velocity magnitude? The author should add more details.
  • What are the limitations and assumptions of the numerical model?

Round 2

Reviewer 4 Report

The authors have carefully considered my suggestions and I can now recommend the publication.